# miR-197-3p Promotes Osteosarcoma Stemness and Chemoresistance by Inhibiting SPOPL

**DOI:** 10.3390/jcm12031177

**Published:** 2023-02-01

**Authors:** Jingyong Zhang, Shubao Wang, Yang Bai, Aasi Mohammad Ali, Jiewen Deng, Yushi Chen, Yonghui Fu, Ming He

**Affiliations:** 1Department of Orthopedics, Shengjing Hospital of China Medical University, Shenyang 110004, China; 2Department of Pathology, Shengjing Hospital of China Medical University, Shenyang 110004, China; 3Department of Nursing, Shengjing Hospital of China Medical University, Shenyang 110004, China; 4Clinical Medicine, Dalian Medical University, Dalian 116044, China

**Keywords:** miR-197-3p, SPOPL, osteosarcoma, cancer stem cells, stemness, chemoresistance

## Abstract

First-line treatment for osteosarcoma includes chemotherapy and surgery. However, the five-year survival rate of refractory osteosarcoma remains unsatisfactory. Osteosarcoma cancer stem cells, possessing stemness and chemoresistance, are one of the critical causes of poor response to chemotherapy. Elucidating regulatory signaling pathways of osteosarcoma cancer stem cells may provide a rationale for improving regimens against chemoresistant osteosarcoma. Methotrexate (MTX)-resistant osteosarcoma cells were established. microRNA expression profiles were used for detecting differentially expressed microRNA in resistant clones and the parental cells. microRNA target databases were employed to predict potential microRNA and mRNA interactions. Flow cytometry was performed to measure stem cell marker Prominin-1 (CD133)-positive cells. Immunofluorescence staining was applied to detect CD133 expression. miR-197-3p mimic or anti-miR-197-3p stably transfected cells were used to generate xenograft models. In the study, we found that miR-197-3p was increased in MTX-resistant cell lines. Overexpression of miR-197-3p enhanced the expression of cancer stem cell markers CD133, Octamer-binding protein 4 (*OCT4*), Transcription factor SOX-2 (*SOX2*), and Homeobox protein NANOG (*NANOG*), as well as chemoresistance-associated genes ATP-dependent translocase ABCB1 (*ABCB1*) and Broad substrate specificity ATP-binding cassette transporter ABCG2 (*ABCG2*), whereas miR-197-3p knockdown inhibited stemness and recovered sensitivity to MTX. We also classified the tumor suppressor Speckle-type POZ protein-like (*SPOPL*) as a target of miR-197-3p. The miR-197-3p mutation that could not combine *SPOPL* promoter regions was unable to sustain stemness or chemoresistance. Collectively, we discovered miR-197-3p conferred osteosarcoma stemness and chemotherapy resistance by targeting *SPOPL*, prompting promising therapeutic candidates for refractory osteosarcoma treatment.

## 1. Introduction

The most prevalent primary bone malignancy, osteosarcoma, affects children and adolescents primarily and accounts for one-third of all bone sarcomas. The standard practice for osteosarcoma includes pre-operative chemotherapy, surgical resection, and postoperative chemotherapy. Patients are initially treated with the standard chemotherapy regimen of doxorubicin, cisplatin, and methotrexate (MTX) [1]. For localized osteosarcoma (80% of patients), the 5-year survival rate is around 75% at the time of diagnosis. However, the 5-year survival rates for individuals with metastases largely drop to 20% [2]. Over three decades have passed with the unsatisfactory scenario still present. Growing evidence demonstrates that cancer stem cells with chemoresistance may be one of the causes driving poor clinical outcomes. Cancer stem cells exhibit several characteristics that may confer chemoresistance, such as abnormal expression of ATP-binding cassette subfamily proteins, activating DNA repair ability, and hyperactive apoptotic regulatory elements [3]. Specific surface markers of cancer stem cells, such as the stem cell marker Prominin-1 (CD133), Octamer-binding transcription factor 4 (*OCT4*), Transcription factor SOX-2 (*SOX2*), and Homeobox protein NANOG (*NANOG*), are frequently used for classifying cancer stem cells. The underlying mechanisms of cancer-stem-cell-mediated chemoresistance are not thoroughly investigated. Promising treatment targets could be found by improving our knowledge of the molecular profiles, regulatory networks, and signaling pathways of cancer stem cells.

Ample studies have demonstrated that the dysregulation of miRNAs contributes to osteosarcoma initiation, progression, and recurrence through multiple mechanisms, including autophagy activation, apoptosis evasion, stemness maintenance, drug resistance induction, and metastasis promotion [4,5]. Numerous studies have shown that miR-197-3p orchestrates the initiation and progression of multiple malignancies as a double-edged sword. On the one hand, miR-197-3p works as a tumor suppressor. For example, the downregulation of miR-197-3p enhanced the growth and invasiveness of osteosarcoma cells in a Krüppel-like factor-10-dependent manner [6]. Circular RNA circMMD_007 suppressed miR-197-3p expression, thereby promoting lung adenocarcinoma progression [7]. Decreased miR-197-3p expression activated Akt and ERK signaling cascades, maintaining liver cancer malignancy [8]. On the other hand, the diminishment of miR-197-3p by long noncoding RNA tumor suppressor candidate 8 retarded osteosarcoma development [9]. Circular RNA hsa_circ_001988 competently sponged to miR-197-3p, thereby blocking gastric cancer metastasis [10]. Expanding the investigation to chemoresistance and cancer stem cells may help draw a global picture of miR-197-3p-centric networks.

Speckle-type pox virus and zinc finger protein (*SPOP*) is an E3 ubiquitin ligase protein, enabling target protein ubiquitination and following proteasomal degradation. Frequent SPOP mutation was found in various malignancies, such as prostate, breast, lung, gastric, and colorectal cancers [11]. The *SPOP* paralog, Speckle-type pox virus, and Speckle-type POZ protein-like (*SPOPL*) act as homodimers or heterodimers with *SPOP* [11]. *SPOPL* deletion is found in The Cancer Genome Atlas prostate cancers, comparable to *SPOP* deletion in prostate cancer [12]. Furthermore, *SPOPL* expression was reduced in medulloblastoma and correlated with differentiation levels [13]. An increased *SPOPL* expression was associated with favorable triple-negative breast cancer clinical outcomes [14]. The function of *SPOPL* in osteosarcoma is still poorly understood.

In this study, we found elevated miR-197-3p expression in MTX-resistant osteosarcoma cell lines and targeted SPOPL. We attempted to uncover the mechanisms of miR-197-3p-SPOPL-mediated osteosarcoma stemness and chemoresistance.

## 2. Materials and Methods

### 2.1. Cell Culture and Establishment of MTX-Resistant Osteosarcoma Subclones

The human osteoblast hFOB1.19 (catalog: CRL-11372), human osteosarcoma cell lines U2OS (catalog: HTB-96), and MG63 (catalog: CRL-1427) were sourced from the American Type Culture Collection (Manassas, VA, USA). U2OS and MG63 were cultured in Dulbecco’s modified Eagle medium (catalog: BE12-604F, Lonza, Basel, Switzerland) supplemented with 10% fetal bovine serum (catalog: F7524, Sigma-Aldrich, St. Louis, MO, USA). hFOB1.19 cells were cultivated in hFOB1.19 complete medium (catalog: EP-ML-0353, Elabscience, Wuhan, China). All cells were kept at 37 °C in a 5% CO_2_ environment.

Parental U2OS and MG63 cells were initially exposed to 3 ng/mL (0.01 µM) MTX [15]. Subjecting the parental cell lines to stepwise raised MTX concentrations, MTX-resistant subclones were produced.

### 2.2. Chemical and Reagents

Methotrexate (catalog: S1210) was obtained from Selleck Chemicals (Houston, TX, USA). The transfection reagent Lipofectamine (catalog: LMRNA001) was purchased from Thermo Fisher Scientific (Waltham, MA, USA).

### 2.3. Microarray-Based Analysis of microRNA Expression Profiles

MicroRNA (miRNA) were enriched and purified by miRNeasy Micro Kit (catalog: 217084, QIAGEN, Hilden, Germany), followed by cDNA synthesis with miRCURY LNA RT Kit (catalog: 339340, QIAGEN). miRCURY LNA miRNA Custom PCR Panels were then used to investigate differentially expressed miRNA in the parental U2OS cells and MTX-resistant U2OS cells. Data gained from three independent experiments were analyzed by the Gene globe data analysis center (https://geneglobe.qiagen.com/us/analyze, accessed on 26 September 2020). Data published on Gene Expression Omnibus as GSE223857 (https://www.ncbi.nlm.nih.gov/geo/query/acc.cgi?acc=GSE223857, accessed on 26 September 2020).

### 2.4. MicroRNA Synthesis and Quantitative Real-Time PCR

miRNA and mRNA were collected by RNeasy Micro Kit (catalog: 74004, QIAGEN). A total of 300ng RNA was then reverse transcribed to cDNA by ProtoScript^®^ II Reverse Transcriptase (catalog: M0368, New England Biolabs, Ipswich, MA, USA). Quantitative real-time PCR (qRT-PCR) was performed with Luna^®^ Universal One-Step RT-qPCR Kit (catalog: E3005, New England Biolabs). Reactions were conducted with the Applied Biosystems MiniAmp Thermal Cyclers (Thermo Fisher Scientific, Waltham, MA, USA) using the following steps: 40 cycles of 95 °C for 15 s, 60 °C for 10 s, and 72 °C for 20 s. Following the previously discussed method2^−ΔΔCq^ method [16], gene expression was normalized to the GAPDH to examine the relative expression. pGPU6/Neo vector, miRNA mimics, small interfering RNA (siRNA), short-hairpin RNA (shRNA), miR-197-3p mutation, and the primers for detection were generated by GenePharma (Shanghai, China). The primers for gene detection list can be found in Appendix A.

### 2.5. MTT Assay for Cell Viability

MTT assays were performed to measure cell viability. Cells were seeded at 1 × 10^4^/well in 96-well plates with 100 µL medium, followed by exposure to MTX at different doses for 24 h. Cells were then treated for 4 h at 37 °C using media supplemented with 10 µL of 3-(4,5-dimethylthiazol-2-yl)-2,5-diphenyltetrazolium bromide (MTT) solution at a final concentration of 5 mg/mL. The medium was replaced by a 150 µL dimethyl sulfoxide solution, followed by incubation for 10 min. The absorbance at 490 nm was analyzed by a Microplate Reader (Thermo Fisher Scientific, Waltham, MA, USA).

### 2.6. Flow Cytometry Analysis

A total of 6 × 10^5^ cells were plated in six-well Petri dishes prior to analysis. Twenty-four hours later, cells were collected, fixed, and resuspended gently with propidium iodide solution and recombinant anti-CD133 antibody (catalog: ab278053, Abcam, Shanghai, China). The populations of CD133-positive cells were detected with flow cytometry according to the manufacturer’s instruction (FACSCalibur™; B.D. Biosciences, San Jose, CA, USA). Data from the flow cytometry were then analyzed by CellQuest Pro software (version 5.1; B.D. Biosciences).

### 2.7. Transwell Migration Assay

Matrigel-free Transwell migration assay was used to measure the migratory capability of cells. Polycarbonate cell culture inserts were purchased from Thermo Fisher Scientific. A total of 8 × 10^4^ cells in 200 µL serum-free DMEM were seeded in the upper room of each chamber, whereas the lower room was filled with 600 µL DMEM supplemented with 10% fetal bovine serum. Cells in the upper compartments were removed after 18 h of incubation at 37 °C, while the migrating cells in the lower compartments were stained, examined, and counted under a high-powered microscope.

### 2.8. Western Blots and Antibodies

Protein was collected with 100 μL RIPA lysis buffer with a complete protease inhibitors cocktail (catalog: ab65621, Abcam). An amount of 20 µg protein was loaded on SDS-PAGE gel and transferred to nitrocellulose membranes. The membranes were first blocked for 1 h with 5% skimmed milk, then incubated overnight at 4 °C with primary antibodies at a dilution of 1: 1000. The membranes were then rinsed three times with TBS-T and given an additional hour of incubation with HRP-conjugated secondary antibodies at a dilution of 1:5000. The Western Lighting Ultra was used to measure immunoreactivity (Thermo Fisher Scientific). The primary antibodies Anti-P Glycoprotein antibody (catalog: ab216656), Anti-BCRP/ABCG2 antibody (catalog: ab207732), Anti-beta Actin antibody (catalog: ab8226), the secondary antibodies rabbit anti-mouse IgG H&L (HRP) (catalog: ab6728) and goat anti-rabbit IgG H&L (HRP) (catalog: ab6721) were obtained from Abcam.

### 2.9. Immunofluorescence Staining

Cells were seeded on a 12-well culture plate (Thermo Fisher Scientific) at a density of 60% confluence. CD133 antibody (catalog: ab278053, Abcam) and mounting medium with DAPI (catalog: ab104139, Abcam) were applied to immunofluorescence staining according to the manufacturer’s protocol. Images were then collected with Olympus BX63 (Olympus, Shinjuku City, Tokyo, Japan).

### 2.10. Prediction of miRNA and mRNA Interaction

Targetscanhuman Release7.2 (https://www.targetscan.org/vert_72/, accessed on 30 September 2020) [17], TarBaseV.8 (https://dianalab.e-ce.uth.gr/html/diana/web/index.php?r=tarbasev8/index, accessed on 30 September 2020) [18], and ENCORI (https://starbase.sysu.edu.cn/agoClipRNA.php?source=mRNA, accessed on 30 September 2020) [19] were employed to analyze target genes of miR-197-3p. The predicted intersection was shown by Draw Venn Diagram (https://bioinformatics.psb.ugent.be/webtools/Venn/, accessed on 30 September 2020).

### 2.11. Luciferase Reporter Assay

An amount of 4 × 10^4^ cells per well of a 24-well plate were transfected with the indicated combinations of 400 ng luciferase reporter, 100 ng pCMV-miR vector, pCMV-miR-197-3p, and pCMV-miR-197-3p mutation. Luciferase activity was determined with the dual-luciferase reporter assay system post 24 h transfection with the Luciferase Reporter Assay System (Promega, Madison, WI, USA) as per the manufacturer’s guidelines.

### 2.12. Animal Models

The current animal study was approved by the Institutional Animal Care and Use Committee of Shengjing Hospital of China Medical University. Nine four-week-old female nude BALB/c mice (Vital River laboratory animal technology Co., Ltd., Beijing, China) were maintained in specific-pathogen-free conditions. U2OS cells were stably transfected with vector, miR-197-3p, or anti-miR-197-3p. A total of 5 × 10^6^ U2OS cells in each group were then injected into mice flanks subcutaneously. Seven days later, tumor-bearing mice were randomly divided into three groups of three mice each, when the mean tumor volume approached 100 mm^3^. According to the previous study, mice received MTX intraperitoneal injection at 5 mg/kg weekly for 4 weeks [20]. The tumor volumes were measured every four days. The mice were euthanized on day 28. Tumors were removed, weighed, and recorded following the standard protocol.

### 2.13. Statistical Analysis

The data from all experiments were presented as means plus standard deviation. Differences in groups were evaluated by ordinary one-way Analysis of Variance (ANOVA), followed by Dunnett’s multiple comparisons test. *p* < 0.05 indicated statistical significance. Statistical analysis was performed with GraphPad version 9.0 (San Diego, CA, USA).

## 3. Results

### 3.1. Subsection the CD133-Positive Populations Are Enriched in MTX-Resistant Osteosarcoma Clones

An initial dose of 0.01 µM MTX was used to treat the U2OS and MG63 cell lines. Gradually increasing doses of MTX were then used to select MTX-resistant subclones. Twelve weeks later, MTX-resistant subclones were established, namely MTX-resistant U2OS (U2OSr) and MTX-resistant MG63 (MG63r). As shown in Figure 1A, U2OS (U2OSr) and MTX-resistant MG63 (MG63r) exhibited 75-fold and 61-fold IC_50_ values to the corresponding parental cells, respectively. Increasing evidence demonstrates that stemness appears along with chemoresistance in osteosarcoma progression. The CD133 antigen, also known as Prominin 1, is frequently used to isolate cancer stem cells. The CD133-positive (CD133+) populations of MTX-resistant subclones were almost twice as many as those in the parental cells (Figure 1B). The typical stem cell markers in the MTX-resistant subclones, OCT4, SOX2, and NANOG, were significantly increased (Figure 1C). The migration of the MTX-resistant cells altered little, while that of the parental cells remarkably dropped post-MTX treatment (Figure 1D). The results indicated that the MTX-resistant osteosarcoma cells showed stemness and chemoresistance.

### 3.2. miR-197-3p and miR-20a-5p Increase in MTX-Resistant Subclones and Participate in the Regulation of Chemoresistance and Stemness

microRNAs (miRNAs) play multiple roles in regulating the pathogenesis of osteosarcoma, including stemness and chemoresistance. We intend to figure out the differentially expressed miRNAs in MTX-resistant clones and parental cells. The expression of miR-197-3p was upregulated most significantly among several changed miRNAs according to the results of qPCR-based miRNA microarray (Figure 2).

Furthermore, the association between the overall survival and the differentially expressed miRNAs was analyzed by recruiting KMplot, an online survival analysis tool [15]. miR-9-3p, miR-17-5p, miR-20a-5p, miR-124-3p, and miR-197-3p were significantly correlated to the overall survival time (Appendix A). The expression of miR-197-3p, miR-20a-5p, and miR-17-5p in human osteoblast cell hFOB1.19 and osteosarcoma cell lines were further examined. miR-197-3p and miR-20a-5p expression was notably elevated in MTX-resistant cells compared to the corresponding parental cells, while miR-17-5p changed little (Figure 2B). Hence, the efficacy of miR-197-3p and miR-20a-5p on stemness was studied. As shown in Figure 2C and Appendix A, the expression of OCT4 and SOX2 was apparently elevated in the group of miR-197-3p and miR-20a-5p. Interestingly, NANOG expression was upregulated in the miR-197-3p group while hardly escalated in the miR-20a-5p group. miR-197-3p and miR-20a-5p promoted the cell viability post MTX exposure (Figure 2D and Appendix A). The population of CD133+ cells in the cells with miR-197-3p overexpression was significantly amplified, while that in the cells with miR-20a-5p overexpression changed slightly (Figure 2E and Appendix A). Furthermore, miR-197-3p and miR-20a-5p promoted the migration of osteosarcoma cells (Figure 2E and Appendix A).

The results indicated that miR-20a-5p might not be the dominant factor in the maintenance of osteosarcoma stemness and chemoresistance. We therefore selected miR-197-3p for further investigation due to miR-197-3p appearing to have been more significant in the enhancement.

### 3.3. miR-197-3p Is Pivotal for Sustaining Chemoresistance and Stemness of Osteosarcoma

Functional recovery experiments were conducted to explore the role of miR-197-3p in managing chemoresistance and stemness. U2OS and MG63 cells were transfected with vector, miR-197-3p mimics (miR-197-3p), or anti-miR-197-3p, followed by a 24 h exposure to 2 µM MTX. As shown in Figure 3A and Appendix A, a dramatical increase in the expression of the stem cell markers OCT4, SOX2, and NANOG was observed in the miR-197-3p group, whereas the stem cell markers were reduced in the anti-miR-197-3p group. In contrast to the escalation of ATP-dependent translocase ABCB1 (ABCB1) and ATP-binding cassette transporter ABCG2 (ABCG2) by miR-197-3p, ABCB1 and ABCG2 were reduced by anti-miR-197-3p. In addition, miR-197-3p overexpression enhanced cell viability, while miR-197-3p knockdown repressed viability compared to the vector group (Figure 3B and Appendix A). CD133 was increased by miR-197-3p, while CD133 was decreased by anti-miR-197-3p (Figure 3C,D and Appendix A). Moreover, miR-197-3p boosted the migration capacity of osteosarcoma cells. In contrast with the enhancement by miR-197-3p, migration was downregulated by miR-197-3p knockdown (Figure 3E and Appendix A).

Apart from in vivo experiments, miR-197-3p or miR-197-3p knockdown stably transfected cells were used to examine tumorigenicity in vivo. Figure 3F,G show that the average volumes of tumors in the miR-197-3p group were much larger than those in the vector group, while those in the miR-197-3p knockdown group dramatically fell.

Briefly, the results supported that miR-197-3p was required to preserve chemoresistance and stemness of osteosarcoma.

### 3.4. miR-197-3p Sustains Stemness of Osteosarcoma Cells by Targeting SPOPL

miRNAs control message RNA (mRNA) expression at the posttranscriptional level. Abnormal expression of mRNAs is involved in the regulation of osteosarcoma chemoresistance. miRNA–mRNA interaction databases TargetScan, TarBase and ENCORI were employed to discover potential target genes of miR-197-3p. The Venn diagram in Figure 4A shows that miR-197-3p directly targets more than 200 mRNAs.

Previous research suggested three genes, Speckle-Type BTB/POZ Protein-Like (SPOPL), Mitogen-Activated Protein Kinase 1 (MAPK1), and Bone Morphogenetic Protein Receptor Type 2 (BMPR2) were associated with stemness in cancers. However, the relationship between genes and stemness in osteosarcoma remains unknown. Figure 4B displays the predicted binding sites of miR-197-3p and the promoter regions. miR-197-3p mutation (miR-197-3p mut), insufficient to combine the promoter region of mRNAs, was produced. The expression of miR-197-3p and miR-197-3p mut was determined by qPCR (Appendix A). As shown in Figure 4C, the dual luciferase activity of the SPOPL promoter presented most significantly in the miR-197-3p group. In contrast, the miR-197-3p mutation was unable to repress SPOPL expression. The SPOPL expression showed the largest decrease in the three genes (Figure 4D), which was in line with the relative dual luciferase activity changes. RNA interference was then introduced into cells to test the efficacy of the genes in regulating osteosarcoma stemness. Appendix A demonstrated that small interfering RNA against the genes worked effectively. The silence of SPOPL remarkably enhanced the expression of stem cell markers OCT4, SOX2, and NANOG, while the impairment of MAPK1 or BMPR2 displayed little efficacy on the expression of stem cell markers (Figure 4E), indicating that miR-197-3p regulated osteosarcoma stemness through directly targeting SPOPL.

### 3.5. The miR-197-3p-SPOPL Axis Is Essential for Maintaining Osteosarcoma Chemoresistance and Stemness

Cells were transfected with vector, anti-miR-197-3p, or miR-197-3p mut. The expression of stem cell markers in the miR-197-3p mut group altered little compared to that in vector cells. ABCB1 and ABCG2 expression exhibited similar changes (Figure 5A and Appendix A).

miR-197-3p mut also recovered the chemoresistance against MTX (Figure 5B and Appendix A). In contrast with the downregulation of CD133+ populations in the miR-197-3p knockdown group, the CD133+ populations in the miR-197-3p mut group were almost identical to those in the vector group (Figure 5C and Appendix A). As shown in Figure 5D and Appendix A, the migration ability of the cells in the miR-197-3p mut group was equal to that of vector cells. The results suggested that the miR-197-3p-SPOPL axis was essential to facilitate osteosarcoma chemoresistance and stemness.

## 4. Discussion

Cancer stem cells are a subset of tumor cells that may navigate tumorigenesis, progression, metastasis, and relapse. Cancer stem cells are characterized by self-renewal and sphere-forming like normal stem cells [21]. Additionally, an increased expression of drug efflux transporters *ABCB1* and *ABCG2* was observed in osteosarcoma cancer stem cells [21,22]. The MTX-resistant U2OS and MG63 subclones showed an enhanced expression of stem cell markers and ABC proteins, suggesting that the established chemoresistant cell lines offered a solid foundation for further investigation.

miRNAs are small non-coding RNAs that could inhibit gene expression by repressing mRNA at a transcriptional level. Expression of miRNAs varies in different cancers and may play roles as tumor suppressors or oncogenes according to their target genes. A tremendous amount of miRNAs have been identified in osteosarcoma apoptosis, senescence, survival, proliferation, migration, invasion, and cell cycle distribution [5,23], including miR-197-3p and miR-20a-5p. Previous studies have proved that miR-20a-5p suppressed *SDC2* and *KIF26B*, leading to a reduction in the multi-drug resistance of osteosarcoma [24,25]. The current evidence that miR-20a-5p promoted stemness and chemoresistance was opposite to previous research. Moreover, ectopic expression of miR-20a-5p impeded *Fas* expression and promoted lung metastasis of osteosarcoma in vivo [26], suggesting that miR-20a-5p-mediated regulatory networks are not fully understood. Due to miR-197-3p exhibiting a more significant alteration in MTX-resistant cells than the parental cells, we selected miR-197-3p to explore further.

The mechanisms by which miR-197-3p determines chemoresistance are highly controversial. On the one hand, miR-197-3p, as a tumor suppressor, promotes sensitivity to chemotherapeutic medicine. The knockdown of circular RNA circ_0014130 triggered miR-197-3p expression, enhancing the sensitivity to 5-fluoropyrimidine in colorectal cancer [27]. miR-197-3p targeted thymidylate synthase, thereby attenuating thymidylate synthase-mediated 5-fluoropyrimidine resistance in colon and gastric cancer [28]. Increased miR-197-3p impaired the resistance to 5-fluoropyrimidine in colorectal cancer [29]. On the other hand, the increase in miR-197-3p promoted breast cancer resistance against tamoxifen, a selective estrogen receptor inhibitor [30]. miR-197-3p overexpression augmented the oxaliplatin resistance of colon carcinoma by inhibiting DKC1 [31]. The current study sheds light on the mechanisms by which miR-197-3p conferred osteosarcoma chemoresistance and stemness, filling the gap in knowledge on miR-197-3p-centric networks in osteosarcoma.

Besides modulating chemoresistance, miR-197-3p targeted Interleukin (IL)-6 and suppressed IL-6/JAK/STAT3 pathways, counteracting the selective proteasome inhibitor Bortezomib’s resistance to multiple myeloma [32]. Decreased miR-197-3p enhanced multi-target inhibitor imatinib’s resistance of gastrointestinal stromal tumors by forming a complex with circular RNA circ-CCS and autophagy-related protein 10 [33]. Our research may provide a clue for exploring miR-197-3p in novel therapies against osteosarcoma.

*SPOP* can impede the stemness of several cancers as a tumor suppressor. For example, *SMAD* interacted with *SPOP* and repressed *SPOP* expression. Blockage of *TGF-β/SMAD* cascade signaling boosted *SPOP* expression and inhibited prostate cancer cell stemness [34]. Apart from *TGF-β/SMAD-SPOP* signaling, the *AMPK-BRAF* axis phosphorylated *NANOG* at Ser68, preventing *SPOP*-mediated *NANOG* polyubiquitin and subsequent degradation. As a result, prostate cancer stemness was elevated [35]. In addition, miR-372/373 directly targeted *SPOP* and led to poor differentiation of colorectal cancer [36]. Although much knowledge of SPOP in regulating tumor stemness has been gained, studies of SPOPL in controlling stemness are limited. The present study revealed that miR-197-3p significantly downregulated SPOPL expression by binding the promoter of *SPOPL*, maintaining osteosarcoma stemness and chemoresistance.

There are some restrictions on our research. First, the differentially expressed miRNAs were determined in cell lines. The distribution and expression of the miRNAs in patients with osteosarcoma could be distinct. Collecting the clinical and pathological features of osteosarcoma patients who received neoadjuvant or adjuvant chemotherapy is valuable. Second, the xenograft models were generated with immortal cell lines. Patient-derived xenografts may accurately capture the molecular profiles of osteosarcoma. Other than miR-197-3p-mediated suppression, different signaling pathways may be involved in the modulation of SPOPL. Further investigation on stemness-related pathways could be fascinating.

In summary, the study suggested that miR-197-3p was essential in possessing osteosarcoma stemness and chemotherapy resistance by targeting tumor suppressor SPOPL. Therefore, the current research facilitates the comprehension of osteosarcoma cancer stem cells. The future study aims to clarify the crosstalk between miR-197-3p-centric networks and other cancer-stem cell-mediated pathways, such as cancer-stem-cell-mediated metastasis and recurrence, which would provide promising targets for treating patients with refractory osteosarcoma.

## Figures and Tables

**Figure 1 jcm-12-01177-f001:**
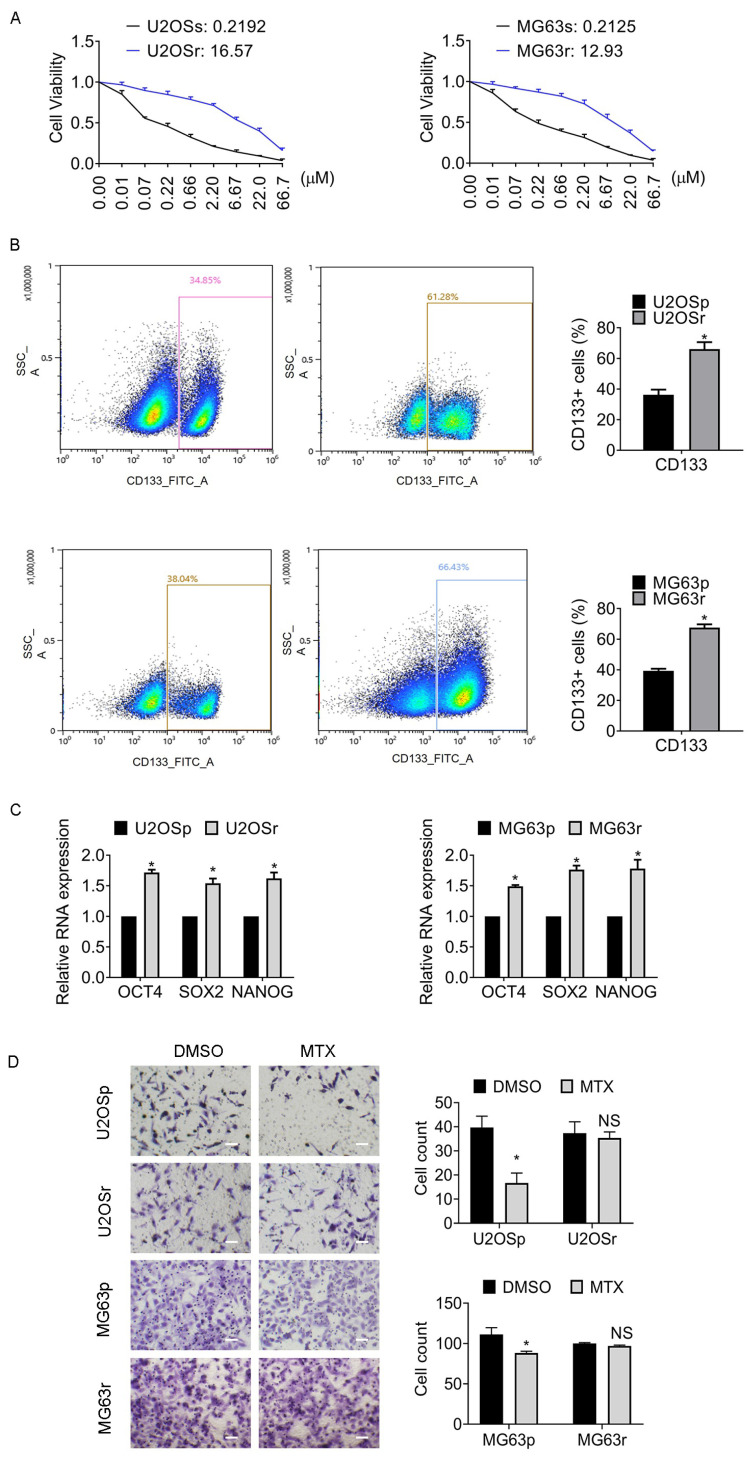
Stem cell marker Prominin-1-positive (CD133+) cells are enriched in methotrexate (MTX)-resistant osteosarcoma subclones. (**A**) IC_50_ values of the indicated clones were determined by cell viability assay. The cells were treated with MTX at different doses for 24 h. (**B**) The representative images of CD133+ cells are shown. The numbers of CD133+ cells were assessed by flow cytometry. SSC-A indicates Side-Scatter Area. (**C**) The expression of stem cell markers Octamer-binding protein 4 (*OCT4*), Transcription factor SOX-2 (*SOX2*), and Homeobox protein NANOG *(NANOG*) was determined by qRT-PCR assay. (**D**) The migration of the cells post 2 µM MTX exposure after 24 h was assessed by transwell assay. Magnification, 200×. Results represented the mean ± S.D. of three independent experiments. * *p* < 0.05; NS, no significance. U2OSp, the parental U2OS; U2Osr, MTX-resistant U2OS subclones; MG63p, the parental MG63; MG63r, MTX-resistant MG63 subclones.

**Figure 2 jcm-12-01177-f002:**
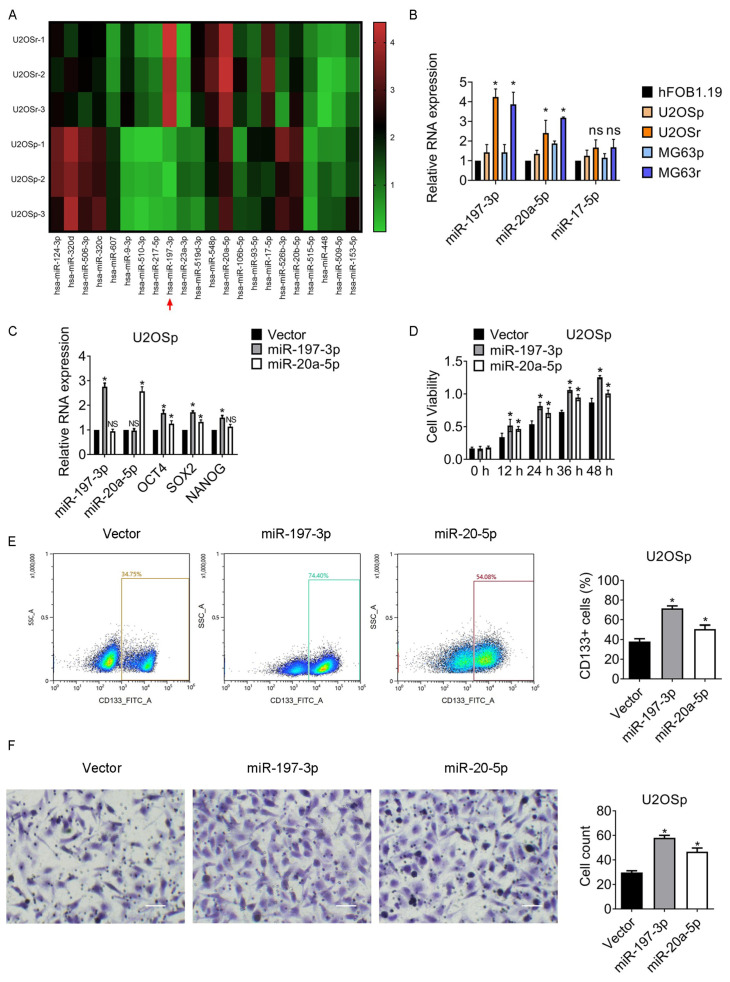
miR-197-3p and miR-20a-5p promote chemoresistance and stemness of MTX-resistant osteosarcoma cells. (**A**) Differentially expressed miRNAs in the indicated cell lines were assessed by qPCR-based miRNA arrays. (**B**) The indicated miRNAs expression in the cell lines was assessed by qPCR assay. Cells were transfected with vector, miR-197-3p, or miR-20a-5p. After 24 h, the cells were treated with 2 µM MTX for 24 h. The following experiments were then conducted. (**C**) The indicated miRNAs and the stem cell markers expression in the cells were assessed by qPCR assay. (**D**) The viability of the indicated cells was measured using MTT assays. (**E**) The representative images of CD133+ cells are shown. The quantity of CD133+ cells was then measured using flow cytometry. SSC-A indicates Side-Scatter Area. (**F**) The migration of the cells was assessed by transwell assay. Magnification, 200×. * *p* < 0.05; NS, no significance. U2OSp, the parental U2OS; U2OSr, MTX-resistant U2OS subclones; MG63p, the parental MG63; MG63r, MTX-resistant MG63 subclones.

**Figure 3 jcm-12-01177-f003:**
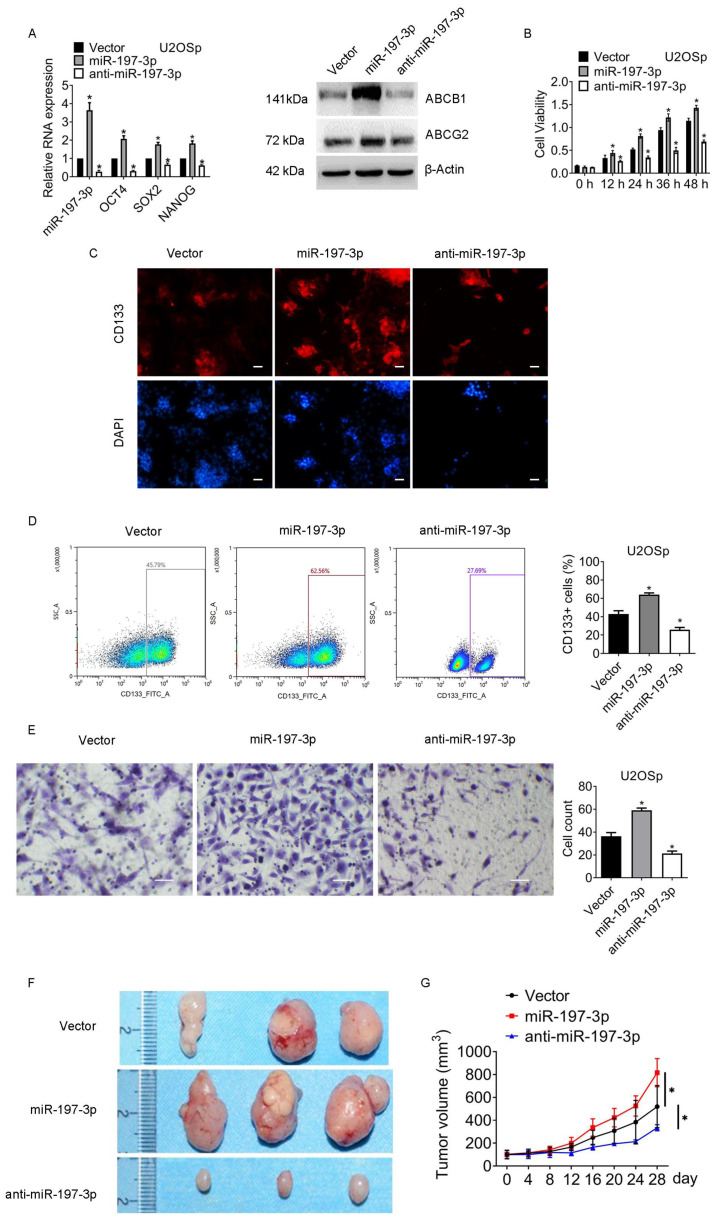
miR-197-3p plays an essential role in maintaining chemoresistance and stemness of osteosarcoma. Cells were transfected with vector, miR-197-3p, or anti-miR-197-3p. After 24 h, the cells were treated with 2 µM MTX for 24 h. The following in vitro experiments were then conducted: (**A**) The indicated genes expression was examined by qPCR and western blot. (**B**) The viability of the cells was assessed by MTT assay. (**C**) Representative images of immunofluorescence staining for CD133 are shown. Magnification, 400×. (**D**) The representative images of CD133+ cells are shown. The quantity of CD133+ cells was measured by flow cytometry. SSC-A indicates Side-Scatter Area. (**E**) The migration of the cells was assessed by transwell assay. Magnification, 200×. (**F**,**G**) Cells were stably transfected with vector, miR-197-3p, or anti-miR-197-3p. Mice received MTX treatment once a week for four weeks. Tumor volumes were calculated every four days. * *p* < 0.05; NS, no significance. Results represented the mean ± S.D. of three independent experiments. U2OSp, the parental U2OS.

**Figure 4 jcm-12-01177-f004:**
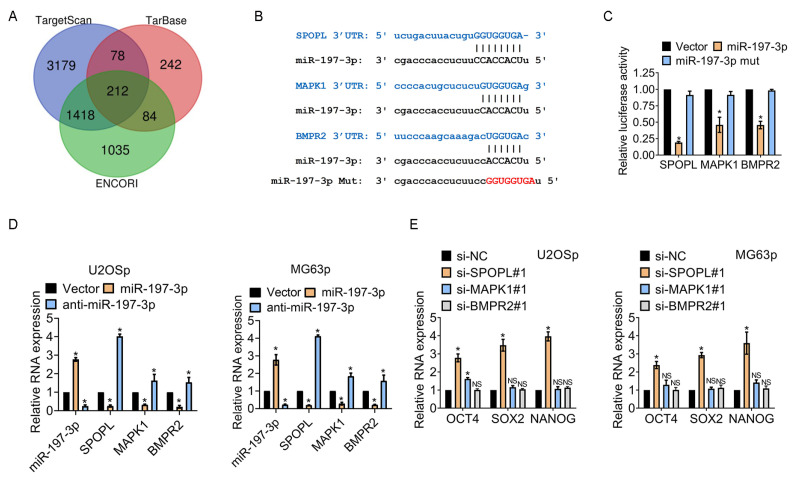
miR-197-3p maintains the stemness of osteosarcoma cells by targeting SPOPL. (**A**) The Venn diagram shows the predicted target genes of miR-197-3p. (**B**) The schematic shows the binding sites of miRNA and potential targets. (**C**) Cells were transfected with vector, miR-197-3p, or miR-197-3p mut. The promoter activity of the indicated genes was determined by dual luciferase assay. Mut indicates mutation. (**D**) The indicated gene expression was assessed by qPCR assay. (**E**) The indicated gene expression was assessed by qPCR assay. * *p* < 0.05; NS, no significance. Results represented the mean ± S.D. of three independent experiments. U2OSp, the parental U2OS.

**Figure 5 jcm-12-01177-f005:**
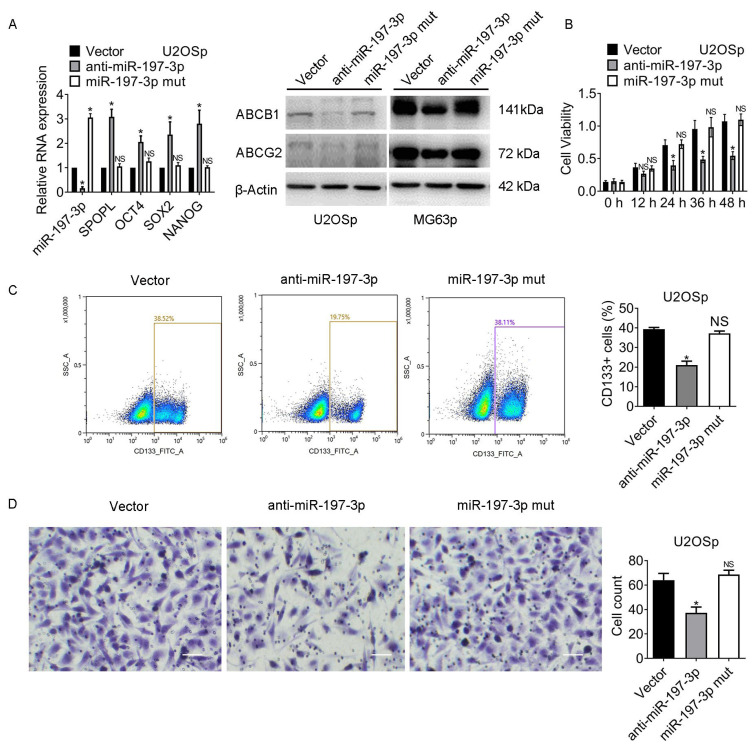
The miR-197-3p-SPOPL axis is crucial for preserving osteosarcoma stemness and chemoresistance. Cells were transfected with vector, anti-miR-197-3p, or miR-197-3p. After 24 h, the cells were exposed to 2 µM MTX for 24 h. The following experiments were then performed: (**A**) The indicated gene expression in the cells was detected by qRT-PCR and Western blot. (**B**) The viability of the cells was assessed by MTT assay. (**C**) The representative images of CD133+ cells are shown. The CD133+ cell population was measured by flow cytometry. SSC-A indicates Side-Scatter Area. (**D**) The migration of the cells was assessed by transwell assay. Magnification, 200×. * *p* < 0.05; NS, no significance. Results represented the mean ± S.D. of three independent experiments. U2OSp, the parental U2OS. MG63p, the parental MG63.

## Data Availability

Data obtained from the miRNA microarray analysis is available on the Gene Expression Omnibus (https://www.ncbi.nlm.nih.gov/geo/query/acc.cgi?acc=GSE223857, accessed on 26 September 2020).

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
