# Peer review of "miR-197-3p Promotes Osteosarcoma Stemness and Chemoresistance by Inhibiting SPOPL"

_jcm, 2023, doi:10.3390/jcm12031177_

Round 1
Reviewer 1 Report
1. The experiments performed in the manuscript titled “miR-197-3p promotes osteosarcoma stemness and chemoresistance by inhibiting SPOPL” looking promising and the findings are seeming to be fruitful. There are few corrections an d justifications are still needed. Following should be clearly discussed before its acceptance.
2. Line 64-68: “Mounting evidence has shown that microRNAs (miRNAs) control cancer development. The dysregulation of miRNAs is associated with tumorigenesis, progression, and recurrence. In a recent study, miRNAs that affected osteosarcoma chemoresistance were divided into five categories: DNA damage response, apoptosis evasion, autophagy activation, abnormal regulation of signaling pathways, and preservation of cancer stem cells”. There must some more recent references other than reference [4].
3. A number of research works related to miR-197-3p have been done. Its application is not limited osteosarcoma only for example, circFBXW7 sponges miR-197-3p and encodes the FBXW7-185aa protein to suppress Triple-Negative Breast Cancer progression though upregulating the FBXW7 expression. Therefore, authors should also discuss its merit/demerit and other applications during in the introduction of miR-197-3p.
4. The unit of time should as “h” rather than “hour/hours”, similarly correct the symbol of degree centigrade throughout the manuscript.
5. What was the doubling time of the cell lines used during MTT assay? What was the incubation time during this study? It was not mentioned in the methodology.
6. As shown in Figure 1. CD133-positive cells are enriched in methotrexate-resistant osteosarcoma subclones, where Figure 1(A) show IC50 values obtained through cell viability assay, at what time point this IC50 values were obtained? Why the Y-axis value in Figure 1A, reaches upto 1.5% only?
7. Mention the molecular weight of the protein (in Figure 5A) used in the Western blot assay
8. A separate section for the conclusions should be mentioned at the end with few sentences about the future prospect of the present findings.
Author Response
- The experiments performed in the manuscript titled “miR-197-3p promotes osteosarcoma stemness and chemoresistance by inhibiting SPOPL” looking promising and the findings are seeming to be fruitful. There are few corrections and justifications are still needed. Following should be clearly discussed before its acceptance.
Response
Thank you for providing us with the opportunity to revise our manuscript. We are grateful to you for the insightful comments on our paper.
We addressed each question and the changes we made immediately after the letter. We corrected the inappropriate labels in Figure 1A and marked the molecular weight on the Western blot images (Figure 5A). In addition, we introduced the studies of miR-197-3p with updated references. The changes are marked in blue on the paper.
All authors have approved the revision of the manuscript. We hope the revised paper fits the Journal of Clinical Medicine's aims and scope better.
Best regards,
Ming He
- Line 64-68: “Mounting evidence has shown that microRNAs (miRNAs) control cancer development. The dysregulation of miRNAs is associated with tumorigenesis, progression, and recurrence. In a recent study, miRNAs that affected osteosarcoma chemoresistance were divided into five categories: DNA damage response, apoptosis evasion, autophagy activation, abnormal regulation of signaling pathways, and preservation of cancer stem cells”. There must some more recent references other than reference [4].
Response
We cited recent studies for introducing the roles of miRNAs in regulating osteosarcoma development as below.
Ample studies have demonstrated that the dysregulation of miRNAs contributes to osteosarcoma initiation, progression, and recurrence through multiple mechanisms, including autophagy activation, apoptosis evasion, stemness maintenance, drug resistance induction, and metastasis promotion. [1,2]
- A number of research works related to miR-197-3p have been done. Its application is not limited osteosarcoma only for example, circFBXW7 sponges miR-197-3p and encodes the FBXW7-185aa protein to suppress Triple-Negative Breast Cancer progression though upregulating the FBXW7 expression. Therefore, authors should also discuss its merit/demerit and other applications during in the introduction of miR-197-3p.
Response
We summarized up-to-date studies of miR-197-3p and explained the need for further exploration. The below context was inserted into the Introduction section.
Numerous studies have shown that miR-197-3p orchestrates multiple malignancies initiation and progression as a double-edged sword. On the one hand, miR-197-3p works as a tumor suppressor. For example, the downregulation of miR-197-3p enhanced the growth and invasiveness of osteosarcoma cells in a Krüppel-like factor 10-dependent manner. [3] circular RNA circMMD_007 suppressed miR-197-3p expression, thereby promoting lung adenocarcinoma progression. [4] Decreased miR-197-3p expression activated Akt and ERK signaling cascades, maintaining liver cancer malignancy. [5] On the other hand, the diminishment of miR-197-3p by long noncoding RNA tumor suppressor candidate 8 retarded osteosarcoma development. [6] Circular RNA hsa_circ_001988 competently sponged to miR-197-3p, thereby blocking gastric cancer metastasis. [7] Expanding the investigation to chemoresistance and cancer stem cells may help draw a global picture of miR-197-3p-centric networks.
- The unit of time should as “h” rather than “hour/hours”, similarly correct the symbol of degree centigrade throughout the manuscript.
Response
All “hour/hours” have been replaced with “h” and all “degree centigrade” has been replaced with °C.
- What was the doubling time of the cell lines used during MTT assay? What was the incubation time during this study? It was not mentioned in the methodology.
Response
The doubling time of hFOB1.19, U2OS, and MG63 cells in the current study was 39.5 h, 25.5 h, and 29.3 h, respectively, consistent with previous studies. [8-10] In addition, we described the details of each experiment where cells were exposed to MTX as below and marked the corresponding context in blue in the revised paper.
- Materials and methods
2.5 MTT assay for cell viability
MTT assays were performed to measure cell viability. Cells were seeded at 1x104/well in 96-well plates with 100 µl medium, followed by an exposure to MTX at different doses for 24 h. Cells were then treated for 4 h at 37 °C using media supplemented with 10 µl of 3-(4,5-dimethylthiazol-2-yl)-2,5-diphenyltetrazolium bromide (MTT) solution at a final concentration of 5 mg/ml.
Figure legends
Figure 1. stem cell marker Prominin-1 positive (CD133+) cells are enriched in methotrexate (MTX)-resistant osteosarcoma subclones.
(A) IC50 values of the indicated clones were determined by cell viability assay. The cells were treated with MTX at different doses for 24 h.
(D) The migration of the cells posts 2 µM MTX exposure 24 h was assessed by transwell assay.
Figure 2. miR-197-3p and miR-20a-5p promote chemoresistance and stemness of MTX-resistant osteosarcoma cells.
(B) The indicated miRNA expression in the cell lines was assessed by qPCR assay. Cells were transfected with vector, miR-197-3p or miR-20a-5p. 24 h later, the cells were treated with 2 µM MTX for 24 h.
Figure 3. miR-197-3p plays an essential role in maintaining chemoresistance and stemness of osteosarcoma. Cells were transfected with vector, miR-197-3p or anti-miR-197-3p. 24 h later, the cells were treated with 2 µM MTX for 24h. The following in vitro experiments were then conducted.
(F and G) Cells were stably transfected with vector, miR-197-3p or anti-miR-197-3p. Mice received MTX treatment once a week for four weeks.
Figure 5. The miR-197-3p-SPOPL axis is crucial for preserving osteosarcoma stemness and chemoresistance. Cells were transfected with vector, anti-miR-197-3p or miR-197-3p. 24 h later, the cells were exposed to 2 µM MTX for 24 h. The following experiments were then performed.
As shown in Figure 1. CD133-positive cells are enriched in methotrexate-resistant osteosarcoma subclones, where Figure 1(A) show IC values obtained through cell viability assay, at what time point this IC50 values were obtained? Why the Y-axis value in Figure 1A, reaches upto 1.5% only?
Response
The cells were treated with MTX at different doses for 24 h. The values of IC50 were then calculated according to the cell viability assay.
Cell viability is measured on a scale of 0 to 1, or alternatively, 0 to 100%. We corrected the inappropriate label in Figure 1A by deleting “%”.
- Mention the molecular weight of the protein (in Figure 5A) used in the Western blot assay.
Response
We have marked the molecular weight of the protein at the proper position (Figures 3A and 5A).
- A separate section for the conclusions should be mentioned at the end with few sentences about the future prospect of the present findings.
Response
We revised the conclusion section as below.
In summary, the study suggested that miR-197-3p was essential in possessing osteosarcoma stemness and chemotherapy resistance by targeting tumor suppressor SPOPL. Therefore, the current research facilitates the comprehension of osteosarcoma cancer stem cells. The future study aims to clarify the crosstalk between miR-197-3p-centric networks and other cancer stem cell-mediated pathways, such as cancer stem cell-mediated metastasis and recurrence, which would provide promising targets for treating patients with refractory osteosarcoma.
References
- Garcia-Ortega, D.Y.; Cabrera-Nieto, S.A.; Caro-Sanchez, H.S.; Cruz-Ramos, M. An overview of resistance to chemotherapy in osteosarcoma and future perspectives. Cancer Drug Resist 2022, 5, 762-793.
- Llobat, L.; Gourbault, O. Role of micrornas in human osteosarcoma: Future perspectives. Biomedicines 2021, 9.
- Wang, L.; Du, Z.G.; Huang, H.; Li, F.S.; Li, G.S.; Xu, S.N. Circ-0003998 promotes cell proliferative ability and invasiveness by binding to mir-197-3p in osteosarcoma. Eur Rev Med Pharmacol Sci 2019, 23, 10638-10646.
- Zhu, L.; Guo, T.; Chen, W.; Lin, Z.; Ye, M.; Pan, X. Circmmd_007 promotes oncogenic effects in the progression of lung adenocarcinoma through microrna-197-3p/protein tyrosine phosphatase non-receptor type 9 axis. Bioengineered 2022, 13, 4991-5004.
- Bi, J.; Guo, Y.; Li, Q.; Liu, L.; Bao, S.; Xu, P. Role of long intergenic non-protein coding rna 01857 in hepatocellular carcinoma malignancy via the regulation of the microrna-197-3p/anterior gradient 2 axis. PLoS One 2021, 16, e0258312.
- Fan, H.; Liu, T.; Tian, H.; Zhang, S. Tusc8 inhibits the development of osteosarcoma by sponging mir‑197‑3p and targeting ehd2. Int J Mol Med 2020, 46, 1311-1320.
- Sun, D.; Wang, G.; Xiao, C.; Xin, Y. Hsa_circ_001988 attenuates gc progression in vitro and in vivo via sponging mir-197-3p. J Cell Physiol 2021, 236, 612-624.
- Solly, K.; Wang, X.; Xu, X.; Strulovici, B.; Zheng, W. Application of real-time cell electronic sensing (rt-ces) technology to cell-based assays. Assay Drug Dev Technol 2004, 2, 363-372.
- Su, Y.; Luo, X.; He, B.C.; Wang, Y.; Chen, L.; Zuo, G.W.; Liu, B.; Bi, Y.; Huang, J.; Zhu, G.H., et al. Establishment and characterization of a new highly metastatic human osteosarcoma cell line. Clin Exp Metastasis 2009, 26, 599-610.
- Harris, S.A.; Enger, R.J.; Riggs, B.L.; Spelsberg, T.C. Development and characterization of a conditionally immortalized human fetal osteoblastic cell line. J Bone Miner Res 1995, 10, 178-186.

Reviewer 2 Report
Comments to the Author
The authors investigate the overexpression of miR-197-3p and its role on osteosarcoma stemness and chemotherapy resistance. The paper is interesting and the results reported are attractive for the possible refractory osteosarcoma treatment, however there are several open questions and a number of changes which are needed. Following are reported some suggestions for the authors. It must also be revised for English fluency.
Abstract:
-Acronyms are defined at first usage in the abstract or the text paper. If a word or phrase is used only once in abstract, text or supporting information sections then an acronym is not used.
-Acronyms not defined in Abstract section: Please specify acronyms
Materials and Methods
Another important question that must be into account; it would be helpful for readers if the U2OS and MG63 cells is described in sufficient detail. For instance, also reporting the differentiation and malignancy levels.
Results
-The figure 1 results are confusing and should be changed, create great confusion and makes the reading of the manuscript difficult. weeks later. MTX-resistant clones of human osteosarcoma cells U2OS and MG63 were established by originally treating the cells with 0.01 µM MTX. Stepwise increased MTX doses were then used to select MTX-resistant subclones. MTX-resistant subclones were generated 12…..as shown in Figure 1A,…..
-The figure 2 results are reported in confusing way and should be changed, create great confusion and makes the reading of the manuscript difficult.
-The figure 3 results are reported in confusing way and should be changed, create great confusion and makes the reading of the manuscript difficult.
-Please place a space between the legend of figure 4 and the results.
Please place a space between the legend of figure 5 and the results.
Author Response
Comments to the Author
The authors investigate the overexpression of miR-197-3p and its role on osteosarcoma stemness and chemotherapy resistance. The paper is interesting and the results reported are attractive for the possible refractory osteosarcoma treatment, however there are several open questions and a number of changes which are needed. Following are reported some suggestions for the authors. It must also be revised for English fluency.
Response
We appreciate your dedicated time and effort in offering creative feedback on our manuscript. Your suggestions greatly help improve the quality of the article.
We responded to the concerns point-by-point as below and revised the language of the paper thoroughly, including correcting spelling and grammatical errors and rewriting blurry sentences.
All authors have approved the revision of the manuscript. We hope the revised paper fits the Journal of Clinical Medicine's aims and scope better.
Best regards,
Ming He
Abstract:
-Acronyms are defined at first usage in the abstract or the text paper. If a word or phrase is used only once in abstract, text or supporting information sections then an acronym is not used.
-Acronyms not defined in Abstract section: Please specify acronyms.
Response
We defined acronyms when they first appeared in the abstract, the main text, and the figure legend sections according to the author guidelines of Journal of Clinical Medicine.
The acronyms include CD133, OCT4, SOX2, NANOG, and MTX.
Materials and Methods
Another important question that must be into account; it would be helpful for readers if the U2OS and MG63 cells is described in sufficient detail. For instance, also reporting the differentiation and malignancy levels.
Response
The origins of human osteosarcoma U2OS cells were taken from a moderately differentiated tibia sarcoma, possessing epithelial adherent morphology [1]. U2OS cells are positive for cartilage markers but negative for nearly all osteoblastic markers. In particular, U2OS cells expressed type IV collagen, which appears in the early stages [2,3]. The human osteoblast-like MG63 cells exhibit fibroblast morphology and produce high interferon yields [4]. The MG63 cells displayed both mature and immature osteoblastic characters, showing a high differentiation potential in response to multiple interventions [5]. Therefore, U2OS and MG63 cells are commonly used for multiple research areas, including arthritis, bone malignancies, and bone development [6].
The doubling time of U2OS and MG63 cells in the current study were 25.5h and 29.3h, respectively, consistent with previous studies [7,8].
Results
-The figure 1 results are confusing and should be changed, create great confusion and makes the reading of the manuscript difficult. weeks later. MTX-resistant clones of human osteosarcoma cells U2OS and MG63 were established by originally treating the cells with 0.01 µM MTX. Stepwise increased MTX doses were then used to select MTX-resistant subclones.MTX-resistant subclones were generated 12…..as shown in Figure 1A,…..
Response
We rewrote the results in Figure 1 as below.
An initial dose of 0.01 µM MTX was used to treat the U2OS and MG63 cell lines. Gradually increasing doses of MTX were then used to select MTX-resistant subclones. Twelve weeks later, MTX-resistant subclones were established, namely MTX-resistant U2OS (U2OSr) and MTX-resistant MG63 (MG63r).
-The figure 2 results are reported in confusing way and should be changed, create great confusion and makes the reading of the manuscript difficult.
Response
We described the details of the results in Figure 2 and improved some blurry sentences as below.
microRNAs (miRNAs) play multiple roles in regulating the pathogenesis of osteosarcoma, including stemness and chemoresistance. We intend to determine the differentially expressed miRNAs in MTX-resistant clones and parental cells. The expression of miR-197-3p was upregulated most significantly among several changed miRNAs according to the results of qPCR-based miRNA microarray (Figure 2).
Besides, the association between the overall survival and the differentially expressed miRNAs was analyzed by recruiting KMplot, an online survival analysis tool.[15] miR-9-3p, miR-17-5p, miR-20a-5p, miR-124-3p, and miR-197-3p were significantly correlated to the overall survival time (Supplementary Table S1). The expression of miR-197-3p, miR-20a-5p, and miR-17-5p in human osteoblast cell hFOB1.19 and osteosarcoma cell lines were further examined. miR-197-3p and miR-20a-5p expression were notably elevated in MTX-resistant cells compared to the corresponding parental cells, while miR-17-5p changed little (Figure 2B). Hence the efficacy of miR-197-3p and miR-20a-5p on stemness was studied. As shown in Figure 2C and Supplementary Figure 1A, the expression of OCT4 and SOX2 apparently elevated in the group of miR-197-3p and miR-20a-5p. Interestingly, NANOG expression was upregulated in the miR-197-3p group while hardly escalating in the miR-20a-5p group. miR-197-3p and miR-20a-5p promoted the cell viability post MTX exposure (Figure 2D and Supplementary Figure 1B). The population of CD133+ cells in the cells with miR-197-3p overexpression was significantly amplified, while that in the cells with miR-20a-5p overexpression changed slightly (Figure 2E and Supplementary Figure 1C). Furthermore, miR-197-3p and miR-20a-5p promoted the migration of osteosarcoma cells (Figure 2E and Supplementary Figure 1D).
The results indicated that miR-20a-5p might not be the dominant factor in the maintenance of osteosarcoma stemness and chemoresistance. We, therefore, selected miR-197-3p for further investigation due to miR-197-3p appeared to have been more significant in the enhancement.
-The figure 3 results are reported in confusing way and should be changed, create great confusion and makes the reading of the manuscript difficult.
Response
We described the details of the results in Figure 3 and improved some blurry sentences as below.
Gain-of-function and loss-of-function experiments were conducted to explore the role of miR-197-3p in managing chemoresistance and stemness. U2OS and MG63 cells were transfected with vector, miR-197-3p mimics (miR-197-3p) or anti-miR-197-3p, followed by a 24-hour exposure to 2 µM MTX. As shown in Figure 3A and Supplementary Figure 2A, a dramatical increase of the stem cell markers OCT4, SOX2, and NANOG expression was observed in the miR-197-3p group. In contrast, the stem cell markers were reduced in the anti-miR-197-3p group. In contrast to the escalation of ATP-dependent translocase ABCB1 (ABCB1) and ATP-binding cassette transporter ABCG2 (ABCG2) by miR-197-3p, ABCB1 and ABCG2 were reduced by anti-miR-197-3p. In addition, miR-197-3p overexpression enhanced cell viability, while miR-197-3p knockdown repressed viability compared to the vector group (Figure 3B and Supplementary Figure 2B). CD133 was increased by miR-197-3p, while CD133 was decreased by anti-miR-197-3p (Figure 3C, 3D and Supplementary Figure 2C). Moreover, miR-197-3p boosted the migration capacity of osteosarcoma cells. In contrast with the enhancement by miR-197-3p, migration was downregulated by miR-197-3p knockdown (Figure 3E and Supplementary Figure 2D).
Apart from in vivo experiments, miR-197-3p or miR-197-3p knockdown stably transfected cells were used to examine tumorigenicity in vivo. Figures 3F and G show that the average volumes of tumors in the miR-197-3p group were much larger than those in the vector group, while those in the miR-197-3p knockdown group dramatically fell.
Briefly, the results supported that miR-197-3p was required to preserve chemoresistance and stemness of osteosarcoma.
-Please place a space between the legend of figure 4 and the results.
Response
We set an indent between the legend of figure 4 and the results and set different font sizes for figure legend and results sections.
-Please place a space between the legend of figure 5 and the results.
Response
We set an indent between the legend of figure 5 and the results and set different font sizes for figure legend and results sections.
References
- Ponten, J.; Saksela, E. Two established in vitro cell lines from human mesenchymal tumours. Int J Cancer 1967, 2, 434-447.
- Becker, J.; Schuppan, D.; Benzian, H.; Bals, T.; Hahn, E.G.; Cantaluppi, C.; Reichart, P. Immunohistochemical distribution of collagens types iv, v, and vi and of pro-collagens types i and iii in human alveolar bone and dentine. J Histochem Cytochem 1986, 34, 1417-1429.
- Chichester, C.O.; Fernandez, M.; Minguell, J.J. Extracellular matrix gene expression by human bone marrow stroma and by marrow fibroblasts. Cell Adhes Commun 1993, 1, 93-99.
- Billiau, A.; Edy, V.G.; Heremans, H.; Van Damme, J.; Desmyter, J.; Georgiades, J.A.; De Somer, P. Human interferon: Mass production in a newly established cell line, mg-63. Antimicrob Agents Chemother 1977, 12, 11-15.
- Pautke, C.; Schieker, M.; Tischer, T.; Kolk, A.; Neth, P.; Mutschler, W.; Milz, S. Characterization of osteosarcoma cell lines mg-63, saos-2 and u-2 os in comparison to human osteoblasts. Anticancer Res 2004, 24, 3743-3748.
- Lauvrak, S.U.; Munthe, E.; Kresse, S.H.; Stratford, E.W.; Namlos, H.M.; Meza-Zepeda, L.A.; Myklebost, O. Functional characterisation of osteosarcoma cell lines and identification of mrnas and mirnas associated with aggressive cancer phenotypes. Br J Cancer 2013, 109, 2228-2236.
- Solly, K.; Wang, X.; Xu, X.; Strulovici, B.; Zheng, W. Application of real-time cell electronic sensing (rt-ces) technology to cell-based assays. Assay Drug Dev Technol 2004, 2, 363-372.
- Su, Y.; Luo, X.; He, B.C.; Wang, Y.; Chen, L.; Zuo, G.W.; Liu, B.; Bi, Y.; Huang, J.; Zhu, G.H., et al. Establishment and characterization of a new highly metastatic human osteosarcoma cell line. Clin Exp Metastasis 2009, 26, 599-610.

Round 2
Reviewer 1 Report
Manuscript has been revised as per the suggestions.